GROWTH OF A SINKHOLE IN A SEISMIC ZONE OF THE NORTHERN APENNINES (ITALY)

Alessandro La Rosa[1,2], Carolina Pagli[2], Giancarlo Molli[2], Francesco Casu[3], Claudio De Luca[3], Amerino Pieroni[4] and Giacomo D'amato Avanzi[2]

[1] Dipartimento di Scienze della Terra, Università degli Studi di Firenze, Via G. La Pira, 4, 50121 Firenze, Italy

[2] Dipartimento di Scienze della Terra, Università di Pisa, Via S. Maria, 53, 56126 Pisa, Italy

[3] CNR, Consiglio Nazionale delle Ricerche, Istituto per il Rilevamento Elettromagnetico dell'Ambiente (IREA-CNR), Via Diocleziano, 328, 80124 Napoli, Italy

[4] Pro.Geo. s.r.l. Via Valmaira, 14, 55032, Castelnuovo di Garfagnana, Italy

**Keywords:** Sinkhole, InSAR, Seismicity

**Abstract**

Sinkhole collapse is a major hazard causing substantial social and economic losses. However, the surface deformations and sinkhole evolution are rarely recorded, as these sites are known mainly after a collapse, making the assessment of sinkholes-related hazard challenging. Furthermore, more than 40% of the sinkholes of Italy are in seismically hazardous zones; it remains unclear whether seismicity may trigger sinkhole collapse. Here we use a multidisciplinary dataset of InSAR, surface mapping and historical records of sinkhole activity to show that the Prà di Lama lake is a long-lived sinkhole that was formed over a century ago in an active fault zone and grew through several events of unrest characterized by episodic subsidence and lake-level changes. Moreover, InSAR shows that continuous aseismic subsidence at rates of up to 7.1 mm yr$^{-1}$ occurred during 2003-2008, between events of unrest. Earthquakes on the major faults near the sinkhole are not a trigger to sinkhole activity but small-magnitude earthquakes at 4-12 km depth occurred during sinkhole unrest in 1996 and 2016. We interpret our observations as evidence of seismic creep at depth causing fracturing and ultimately leading to the formation and growth of the Prà di Lama sinkhole.

## 1. Introduction

Sinkholes are closed depressions with internal drainage typically associated with karst environments, where the exposed soluble rocks are dissolved by circulating ground water (dissolution sinkholes) but other types of sinkholes also exist. Subsidence sinkholes, for example, can form for both internal erosion and dissolution of covered layers leading to downward gravitational deformations such as collapse, sagging or suffosion (*Ford and Williams, 2007; Gutiérrez et al., 2008*). Deep sinkholes have been often observed along seismically active faults indicating a causal link between sinkhole formation and active tectonics (*Faccenna et al., 1993; Harrison et al., 2002; Closson et al., 2005; Florea, 2005; Del Prete et al., 2010; Parise et al., 2010; Wadas et al., 2017*). In some cases, the processes responsible for their formation have been attributed to fracturing and increased permeability in the fault damage zone promoting fluid circulation and weathering of soluble rocks at depth. Additionally, when carbonate bedrocks lie below thick non-carbonate formations, stress changes caused by faulting may cause decompression of confined aquifers favouring upward migration of deep fluids, hence promoting erosion and collapses (e.g. *Harrison et al., 2002; Wadas et al., 2017*). Seismically-induced stress changes could also trigger collapse of unstable cavities as in the case of the two sinkholes that formed near En Gedi (Dead Sea) following the $M_w$ 5.2 earthquake on the Dead Sea Transform Fault in 2004 (*Salamon, 2004*). Sinkhole subsidence and collapses are a major hazard and cause substantial economic and human losses globally (*Frumkin and Raz, 2001*; *Closson,* 2005; *Wadas,* 2017).

In Italy, a total of 750 sinkholes have been identified and the 40% of them are along active faults (*Caramanna et al.*, 2008) but this number could be underestimated due to the high frequency of sinkholes both related to karst and anthropogenic origin (*Parise and Vennari, 2013*). Seismicity induced sinkhole deformation have been often observed in Italy (*e.g. Santo et al., 2007; Parise et al., 2010; Kawashima et al., 2010*).

The sinkhole of Prà di Lama, near the village of Pieve Fosciana (Lucca province, Italy), is a quasi-circular depression filled by a lake. Prà di Lama is located in the seismically active Apennine range of Northern Tuscany, at the intersection between two active faults (Fig. 1). Hot springs are also present at Pieve Fosciana suggesting that fluid migration along the faults planes occurs. Sudden lake-level changes of up to several meters, ground subsidence, surface fracturing and seismicity have occurred repeatedly since at least 991 A.D. (*Nisio, 2008*). The most recent deformation events occurred in March 1996 and between May 2016 and October 2017. However, the processes that control the growth of the Prà di Lama sinkhole remain unclear. Furthermore, whether seismicity along the active faults around Prà di Lama may trigger sinkhole subsidence or collapse is debated.

In this paper we combine recent InSAR observations, seismicity, and surface mapping, as well as historical records of lake-level changes and ground subsidence at the Prà di Lama from 1828 to understand the mechanisms of sinkhole growth in an active fault system.

## 2. Geological setting

The area of the Prà di Lama sinkhole is located within the Garfagnana basin (Fig.1), an extensional graben in the western Northern Apennines, a NW-SE trending fold-and-thrust belt formed by the stack of different tectonic units caused by the convergence of the Corsica-European and Adria plates. The current tectonic regime of the Apennines is characterized by shortening in the eastern sector of the Apennine range and extension in the westernmost side of the range (*Elter et al., 1975; Patacca and Scandone, 1989; Bennett et al., 2012*). The contemporaneous eastward migration of shortening and upper plate extension are believed to be caused by the roll-back subduction during the counter-clockwise rotation of the Adria plate (*Doglioni, 1991; Meletti et al., 2000; Serpelloni et al., 2005; Faccenna et al., 2014; Le Breton et al., 2017*). Extension started 4-5 Ma ago leading to the formation of several NW-SE-oriented grabens, bounded by NE-dipping and SW-dipping normal faults that are dissected by several NE-trending, right-lateral strike-slip faults (Fig.

1). The inner northern Apennines are a seismically active area, where several earthquakes with $M_W$ > 5 occurred, including the largest instrumentally recorded earthquake, $M_w$ 6.5, in 1920 (*Tertulliani and Maramai, 1998; Rovida et al., 2016; Bonini et al., 2016*) and the most recent $M_w$ 5.1 earthquake in 2013 (*Pezzo et al., 2014; Stramondo et al., 2014; Molli et al., 2016*).

The uppermost stratigraphy at Prà di Lama consists of 8m-thick layer of alluvial and palustrine gravels and sandy deposits containing peaty levels, covering an ~85m-thick sandy-to-silty fluvio-lacustrine deposits with low permeability (from Villafranchian to present age) (*Chetoni, 1995*) (Fig.2a and b). These deposits cover a ~1000m-thick turbiditic sequence (Macigno Fm). Below it, a sequence of carbonate rocks pertaining to the Tuscan Nappe Unit is present reaching down to a depth of ~2000 m, where anhydrites (Burano fm.) and calcareous-dolomitic breccias (Calcare Cavernoso Fm.) overlie the Tuscan Metamorphic Units (Fig. 2c).

The Prà di Lama lake lies at the centre of a depression (Figs. 2 and 5). The low slopes characterizing the topography of the area results in the absence of active gravitational ground motions (Fig 2). Furthermore, the Prà di Lama sinkhole is an isolated feature in the region being the only mapped sinkhole in the entire Garfagnana graben (*Caramanna et al., 2008*); the closest sinkhole is in Camaiore (*Buchignani et al., 2008*) near the Tuscany coast (Fig.1).

The Prà di Lama sinkhole is located at the intersection between two seismically active faults: the Corfino normal fault (*Itacha working group, 2003; Di Naccio et al., 2013; ISIDe working group, 2016*) and the right-lateral strike-slip fault M.Perpoli-T.Scoltenna that recently generated the Mw 4.8 earthquake in January 2013 (Fig.1) (*Vannoli, 2013; Pinelli, 2013; Molli et al., 2017*). Hot water springs are also present at Prà di Lama (*Bencini et al., 1977; Gherardi and Pierotti, 2018*). Geochemical analyses of the Prà di Lama spring waters by *Gherardi and Pierotti (2018)*, expanding on previous research (*Baldacci et al., 2007*), suggest that both shallow and deep aquifers are present below Prà di Lama (Fig. 2b). Shallow aquifers have low salinity and low temperature while waters

feeding the thermal springs have high temperature (~57 °C) and high salinity (5.9g/kgw), suggesting

the presence of a deep aquifer at ~2000 m into the anhydrite and the calcareous-dolomitic breccia.

The high salinity of the deep groundwaters is associated with dissolution of the deep evaporitic

formations. Furthermore, un-mixing of deep and shallow waters is interpreted by *Gherardi and*

*Pierotti (2018)* as an evidence of their rapid upwelling, likely occurring along the existing faults.

**3. Data**

Century-scale historical records of sinkhole activity are available at Prà di Lama and allow us

to determine the timescale of sinkhole evolution as well as to characterize the different events of

unrest, in particular the two most recent events in 1996 and 2016. InSAR time-series analysis is also

carried out to measure ground deformations in the Prà di Lama sinkhole in the time period between

events of unrest. Finally, the local catalogue of seismicity (ISIDE catalogue, INGV) is used to inform

us on the timing and types of brittle failures in the area of the sinkhole.

**3.1 Historical Record**

The first historical record of the Prà di Lama sinkhole dates back to the 991 A.D., when the

area was described as a seasonal shallow pool fed by springs. Since then, the depression grew and

several events of unrest consisting of fracturing and fluctuations of the lake level were reported

(*Raffaelli, 1869; De Stefani, 1879, Giovannetti, 1975*) (Table 1). In particular, eight events of unrest

were reported, giving an average of 1 event of unrest every 26 years. We conducted direct

observation of surface deformation around the lake for the two most recent events in 1996 and

2016.

In 1996, the lake level experienced a fall of up to 4 m (Fig. 3 and Fig. S1) and at the same time

the springs outside the lake suddenly increased the water outflow. Clay and mud were also ejected

by the springs outside the lake while fractures and slumps occurred within the lake due to the water

drop (Fig. 3 and Fig. S1). The unrest lasted approximately 2 months, from March to April 1996.

During the final stages, the water level in the lake rose rapidly, recovering its initial level, and
contemporaneously the springs water flow reduced.
In June 2016, an event of unrest consisting of ground subsidence on the western and southern sides
of the Prà di Lama lake started and lasted approximately 9 months, until February 2017. During this
period fractures formed and progressively grew, increasing their throw to up to 70 cm and affecting
a large area on the western side of the lake (Fig. 3 and Fig. S2). Subsidence around the lake resulted
in an increase of the lake surface, in particular on the western side and in the formation of tensile
fractures (Fig. 3 and Fig. S2). Unlike the 1996 events of unrest, no lake level changes or increase of
water flow from the springs around the lake were observed.
**3.2 InSAR**
InSAR is ideally suited to monitor localized ground deformation such as caused by sinkholes
as it can observe rapidly evolving deformation of the ground at high spatial resolution (*Baer et al.,*
*2002; Castañeda et al., 2009; Atzori et al., 2015; Abelson et al., 2017*). Furthermore, the availability
of relatively long datasets of SAR images in the Apennine allows us to study the behaviour of the
Prà di Lama sinkhole using multi-temporal techniques. We processed a total of 200 interferograms
using SAR images acquired by the ENVISAT satellite between 2003 to 2010 from two distinct tracks
in Ascending or Descending viewing geometry (tracks 215 and 437). We used the Small BAseline
Subset (SBAS) multi-interferogram method originally developed by *Berardino et al. (2002)* and
recently implemented for parallel computing processing (P-SBAS) by *Casu et al. (2014)* to obtain
incremental and cumulative time-series of InSAR Line-of-Sight (LOS) displacements as well as maps
of average LOS velocity. In particular, the InSAR processing has been carried out via the ESA platform
P-SBAS open-access on-line tool named G-POD (Grid Processing On Demand) that allows generating
ground displacement time series from a set of SAR data (*De Luca et al., 2015*).
The P-SBAS G-POD tool allows the user to set some key parameters to tune the InSAR
processing. In this work, we set a maximum perpendicular baseline (spatial baseline) of 400 m and
maximum temporal baseline of 1500 days. The geocoded pixel dimension was set to ~80 m by 80 m
(corresponding to averaging together 20 pixels in range and 4 pixels in azimuth).
We initially set a coherence threshold to 0.8 (0 to 1 for low to high coherence) in order to
select only highly coherent pixels in our interferograms. The 0.8 coherence threshold is used to
select the pixels for the phase unwrapping step that is carried out by the Extended Minimum Cost
Flow (EMCF) algorithm (*Pepe and Lanari, 2006*). By setting high values of this parameter the pixels
in input to the EMCF algorithm are affected by less noise as compared to selecting low values, thus
increasing the quality of the phase unwrapping step itself and reducing the noise in our final velocity
maps and time-series (*De Luca et al., 2015*; *Cignetti et al., 2016*).
We also inspected the series of interferograms and excluded individual interferograms with low
coherence. We identified and discarded 29 noisy interferograms in track 215A and other 11
interferograms in track 437D. Finally, we applied an Atmospheric Phase Screen (APS) filtering to
mitigate further atmospheric disturbances (*Hassen, 2001*). Accordingly, we used a triangular
temporal filter with a width of 400 days to minimize temporal variations shorter than about a year
as we focus on steady deformations rather than seasonal changes. Shorter time interval of 300 days
was also tested but provided more noisy time-series.
The average velocity map and the incremental time-series of deformation obtained with the
P-SBAS method have to be referred to a stable Reference Point. For our analysis, the reference point
was initially set in the city of Massa because GPS measurements from *Bennett et al. (2012)* show
that the surface velocities there are < 1mm $yr^{-1}$; therefore, Massa can be considered stable.
Assuming Massa as reference point, the average velocity map revealed the deformation pattern
around the Prà di Lama lake. We then moved the reference point outside the sinkhole deformation

pattern but close to the village of Pieve Fosciana (Fig. S3a). Selecting a reference point close to our study area rather than in Massa allowed us to better minimize the spatially correlated atmospheric artefacts.

As a final post processing step we also calculated the vertical and east-west components of the velocity field in the area covered by both the ascending and descending tracks and assuming no north-south displacement. Given that the study area is imaged by the ENVISAT satellite from two symmetrical geometries with similar incidence angles (few degrees of difference), the vertical and east-west components of the velocity field can simply be obtained solving the following system of equations (*Manzo et al., 2006*):

$$\begin{cases} v_H = \dfrac{\cos\vartheta}{\sin(2\vartheta)}\,(v_{DESC} - v_{ASC}) = \dfrac{v_{DESC} - v_{ASC}}{2\sin\vartheta} \\ v_V = \dfrac{\sin\vartheta}{\sin(2\vartheta)}\,(v_{DESC} + v_{ASC}) = \dfrac{v_{DESC} + v_{ASC}}{2\cos\vartheta} \end{cases}$$

where $v_H$ and $v_V$ are the horizontal and vertical component of the velocity field, $v_{DESC}$ and $v_{ASC}$ are the average LOS velocities in the Descending and Ascending tracks, respectively; $\vartheta$ is the incidence angle.

The InSAR P-SBAS analysis shows that significant surface deformation occurs at Pieve Fosciana between 2003 and 2010. The observed deformation pattern consists of range increase mainly on the western flank of the Prà di Lama lake. The range increase is observed in both ascending and descending velocity maps (Fig. 4a, b), with average LOS velocities of up to -7.1 mm yr$^{-1}$ decaying to -1 mm yr$^{-1}$ over a distance of 400 m away from the lake. Elsewhere around the lake coherence is not kept due to the presence of both cropland and woodland cover, leading to decorrelation. However, few coherent pixels are identified on the eastern flank of the lake, in areas with buildings and sparse vegetation cover, suggesting that the deformation pattern may be circular, with a radius of ~600 m (Figs. 4 and 5). In order to increase the number of analysed pixels we tested decreasing our coherence threshold from 0.8 to 0.4. The results are displayed in Fig. S3b and show that only a few

more pixels are gained north of the sinkhole as compared to choosing a threshold of 0.8 (Fig. 4). We
conclude that decreasing the coherence threshold does not allow to retrieve the entire deformation
pattern, likely due to the fact the area is vegetated.
The maps of vertical and East-West velocities show vertical rates of -4.6 mm $yr^{-1}$ and horizontal
eastward velocities of 5.4 mm $yr^{-1}$ (Fig. 4c, d) consistent with subsidence and contraction centred at
the lake. Furthermore, figure 5 shows that the current deformation pattern follows the topography,
suggesting that subsidence at Prà di Lama is a long-term feature. The time-series of cumulative LOS
displacements show that subsidence occurred at an approximately constant rate between the 2003
and the 2008 but it slowed down in 2008 (Fig. 4e, f), indicating that subsidence at Prà di Lama occurs
also between events of unrest. Furthermore, our time-series of vertical and east-west cumulative
displacements also confirm that the fastest subsidence and contemporaneous eastward motion
occurred until 2008 (Fig. 4 g, h). In order to better understand the mechanisms responsible for the
sinkhole growth and the different types of episodic unrest we also analysed the seismicity.
**3.3 Seismicity**
Seismicity at the Prà di Lama lake was analysed using the catalogue ISIDe (Italian Seismological
Instrumental and Parametric Data-Base) spanning the time period from 1986 to 2016. We calculated
the cumulative seismic moment release using the relation between seismic moment and
magnitudes given by *Kanamori* (1977). First, we analysed the seismic moment release and the
magnitude content of the earthquakes in the area encompassing the sinkhole and the faults
intersection (10 km radius, Fig. 1) to understand whether unrest at Prà di Lama is triggered by
earthquakes along the active faults (Fig. 6). Figure 6a shows that although several seismic swarms
occurred in the area, no clear temporal correlation between the swarms and the events of unrest
at Prà di Lama is observed, suggesting that the majority of seismic strain released on faults around
the Prà di Lama lake does not affect the activity of the sinkhole. We removed from the plot in figure

6a the large magnitude earthquake, $M_w$ 4.8, on the 25[th] of January, 2013 in order to better visualize the pattern of seismic moment release in time. In any case, no activity at Prà di Lama was reported in January 2013.

We also analysed the local seismicity around the Prà di lama lake, within a circular area of 3 km radius around the lake (Fig. 1), to better understand the deformation processes occurring at the sinkhole and we found that swarms of small-magnitude earthquakes ($M_L \leq 2$) occurred during both events of unrest at Prà di Lama in 1996 and 2016 (Fig. 7a, b, c), while a few earthquakes with magnitudes > 2 occurred irrespective of the events of unrest. This indicates that seismicity during sinkhole activity is characterized by seismic energy released preferentially towards the small end of magnitudes spectrum. This pattern is specific of the sinkhole area as in the broader region (Fig. 6b, c) the majority of earthquakes magnitudes are in the range between $M_L > 2$ and $M_L < 3$ and few $M_L$ > 3 also occurred. We also analysed the hypocentres of the earthquakes around the Prà di lama lake (3 km radius) and find that these range between 4.5 and 11.5 km depth, indicating that deformation processes in the fault zone control the sinkhole activity. On the other hand, no earthquakes were recorded at Prà di Lama during the period of subsidence identified by InSAR between 2003 and 2010, indicating that subsidence between events of unrest continues largely aseismically.

To strengthen our seismicity analysis and clarify whether a connection between major tectonic earthquakes and sinkhole unrest exists, we also analysed the historical parametric seismic catalogues (*Rovida et al., 2016; INGV Catalogo Parametrico dei Terremoti Italiani, CPTI15*). Figure 8 shows the occurrence of major earthquakes, with magnitude > 4.0 up to 20 km distant from Pieve Fosciana and the events of sinkhole unrest at Prà di lama. No clear connection between occurrence of large distant earthquakes and events of sinkhole unrest is observed, suggesting that the mechanisms responsible for activation of the Prà di Lama sinkhole should be attributed to local processes.

4. **Discussion and conclusions**

A multi-disciplinary dataset of InSAR measurements, field observations and seismicity reveal that diverse deformation events occur at the Prà di Lama sinkhole. Two main events of sinkhole unrest occurred at Prà di Lama in 1996 and 2016 but the processes had different features. In 1996 the lake-level dropped together with increased water outflow from the springs, while in 2016 ground subsidence led to the expansion of the lake surface and fracturing. In 2016, fractures formed on the South-Western shore of the lake. The main active strike-slip fault is also oriented SW, suggesting a possible tectonic control on the deformation.

We considered processes not related to the sinkhole activity that could explain the observed deformation at Prà di Lama. Active landslides can cause both vertical and horizontal surface motions (e.g. *Nishiguchi et al., 2017*). However, no landslides are identified in the deforming area around the sinkhole (Fig.3). Furthermore, the low topographic slopes rule out the presence of active landslides in the area. Concentric deformation patterns are observed above shallow aquifers (e.g. *Amelung et al., 1999*). However, deformation caused by aquifers have a seasonal pattern rather than continuous subsidence over the timespan of several years, as in Prà di Lama. A long-term subsidence could only be caused by over-exploitation of an aquifer but no water is pumped from the aquifers in the deforming area around Prà di Lama. We conclude that the observed InSAR deformation is caused by the sinkhole.

InSAR analysis shows that continuous but aseismic subsidence of the sinkhole occurred between the two events of unrest, during the period 2003-2010. Instead swarms of small-magnitude earthquakes coeval to the unrest events of 1996 and 2016 were recorded at depth between 4.5 and 11.5 km, indicating that a link between low magnitude seismicity and sinkhole activity exists. We suggest that seismic creep in the fault zone underneath Prà di Lama occurs, causing the diverse deformation events.

Seismic creep at depth could have induced pressure changes in the aquifer above the fault
zone (1996 events) as well as causing subsidence by increased fracturing (2016 events). The
seismicity pattern revealed by our analysis suggests that the Mt.Perpoli-T.Scoltenna strike-slip fault
system underneath Prà di Lama is locally creeping, producing seismic sequences of low magnitude
earthquakes. Similar seismicity patterns were observed along different active faults (*i.e. Nadeau et al.,*
*1995; Linde et al. 1996; Rau et al., 2007; Chen et al., 2008; Harris, 2017*). In 2006, along the
Superstition Hills fault (San Andreas fault system, California) seismic creep has been favoured by
high water pressure (*Scholz, 1998; Wei et al., 2009; Harris, 2017*). We suggest that along the fault
zone below Prà di Lama an increase in pressure in the aquifer in 1996 caused fracturing at the
bottom of the lake and upward migration of fluids rich in clays, in agreement with the observations
of lake-level drop and mud-rich water ejected by the springs in 1996. Our interpretation is also in
agreement with geochemical data indicating that the high salinity of thermal waters at Prà di Lama
have a deep origin, ~2000 m, where fluid circulation dissolves evaporites and carbonates, creating
cavities and then reaching the surface by rapid upwelling along the faults system *(Gherardi and*
*Pierotti, 2018)*. The presence of deep cavities and a thick non-carbonate sequence suggests that the
Prà di Lama sinkhole is a deep-sited caprock collapse sinkhole according to the sinkhole classification
of *Gutiérrez et al. (2008, 2014)*. Sudden fracturing and periods of compaction of cavities created by
enhanced rock dissolution and upward erosion in the fluid circulation zone could explain both
sudden subsidence and fracturing, as in 2016, and periods of continuous but aseismic subsidence as
in 2003-2010. Similar processes have been envisaged for the formation of a sinkhole at the
Napoleonville Salt Dome, where a seismicity study suggests that fracturing enhanced the rock
permeability, promoting the rising of fluids and, as a consequence, erosion and creation of deep
cavities prone to collapse (*Sibson, 1996; Micklethwaite et al., 2010; Nayak and Dreger, 2014;*
*Yarushina et al., 2017*). Recently, a sequence of seismic events was identified at Mineral Beach
(Dead Sea fault zone) and was interpreted as the result of cracks formation and faulting above
subsurface cavities (*Abelson et al., 2017*).
Precursory subsidence of years to few months has been observed to precede sinkhole collapse
in carbonate or evaporitic bedrocks (e.g. *Baer et al., 2002*; *Nof et al., 2013*; *Cathleen and Bloom,*
*2014; Atzori et al., 201*5; *Abelson et al., 2017*). However, the timing of these processes strongly
depends on the rheological properties of the rocks (*Shalev and Lyakhovsky, 2013*). Furthermore, the
presence of a thick lithoid sequence in Prà di Lama may delay sinkhole collapse, also in agreement
with the exceptionally long timescale (~200 years) of growth of the Prà di Lama sinkhole (*Shalev and*
*Lykovsky, 2012; Abelson et al., 2017*).  However, at present we are not able to establish if and when
a major collapse will occur in Prà di Lama.
We identified a wide range of surface deformation patterns associated with the Prà di Lama
sinkhole and we suggest that a source mechanism for the sinkhole formation and growth is seismic
creep in the active fault zone underneath the sinkhole.  This mechanism could control the evolution
of other active sinkholes in Italy as well as in other areas worldwide where sinkhole form in active
fault systems (e.g. Dead Sea area). InSAR monitoring has already shown to be a valid method to
detect precursory subsidence occurring before a sinkhole collapse and the recent SAR missions, such
as the European Sentinel-1, will very likely provide a powerful tool to identify such deformations.
**Acknowledgements**
We thank the two anonymous reviewers for their constructive and useful comments. We thank the
European Space Agency (ESA) for providing the ENVISAT SAR data used in this study through the
VA4. This work was supported by the ESA G-POD and GEP projects through the Infrastructure of
High Technology for Environmental and Climate Monitoring project for Structural improvement (I-
AMICA-PONa3_00363) financed under the National Operational Programme (NOP) for "Research
and Competitiveness 2007-2013", and co-funded by European Regional Development Fund (ERDF)

and National resources. All processed interferograms are archived at IREA-CNR, Naples. A.L.R. thanks IREA-CNR, Naples for his InSAR-training internship. C.P. gratefully acknowledges the support she received through her Rita Levi Montalcini fellowship (Nota MIUR Montalcini 26259_21/12/2013). The DEM data used in this study are from the SRTM (Shuttle Radar Topography Mission) by JPL (NASA). The Lidar DEM data are from Regione Toscana through GEOscopio webgis portal (http://www502.regione.toscana.it/geoscopio/cartoteca.html). The seismicity data are provided by the Istituto Nazionale di Geofisica e Vulcanologia (INGV) through the Italian Seismological Instrumental and Parametric Data-Base (ISIDe) and the Catalogo Parametrico dei Terremoti Italiani 2015 (CPTI15). This work was also financially supported by Università di Pisa.

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

*Figure 1 - Study area*. The Prà di Lama sinkhole is marked by the yellow star. Black tick lines are faults. Blue dots are the earthquakes between 1986 and 2017. Focal mechanisms are from the Regional Centroid Moment Tensor (RCMT) catalogue. The yellow circles represent the areas with radii of 3km and 10 km used for the seismicity analysis. The red dot is the sinkhole of Camaiore (*Buchignani et al., 2008; Caramanna et al. 2008*). The red box in the *in*set marks the location of the area shown in the main figure.

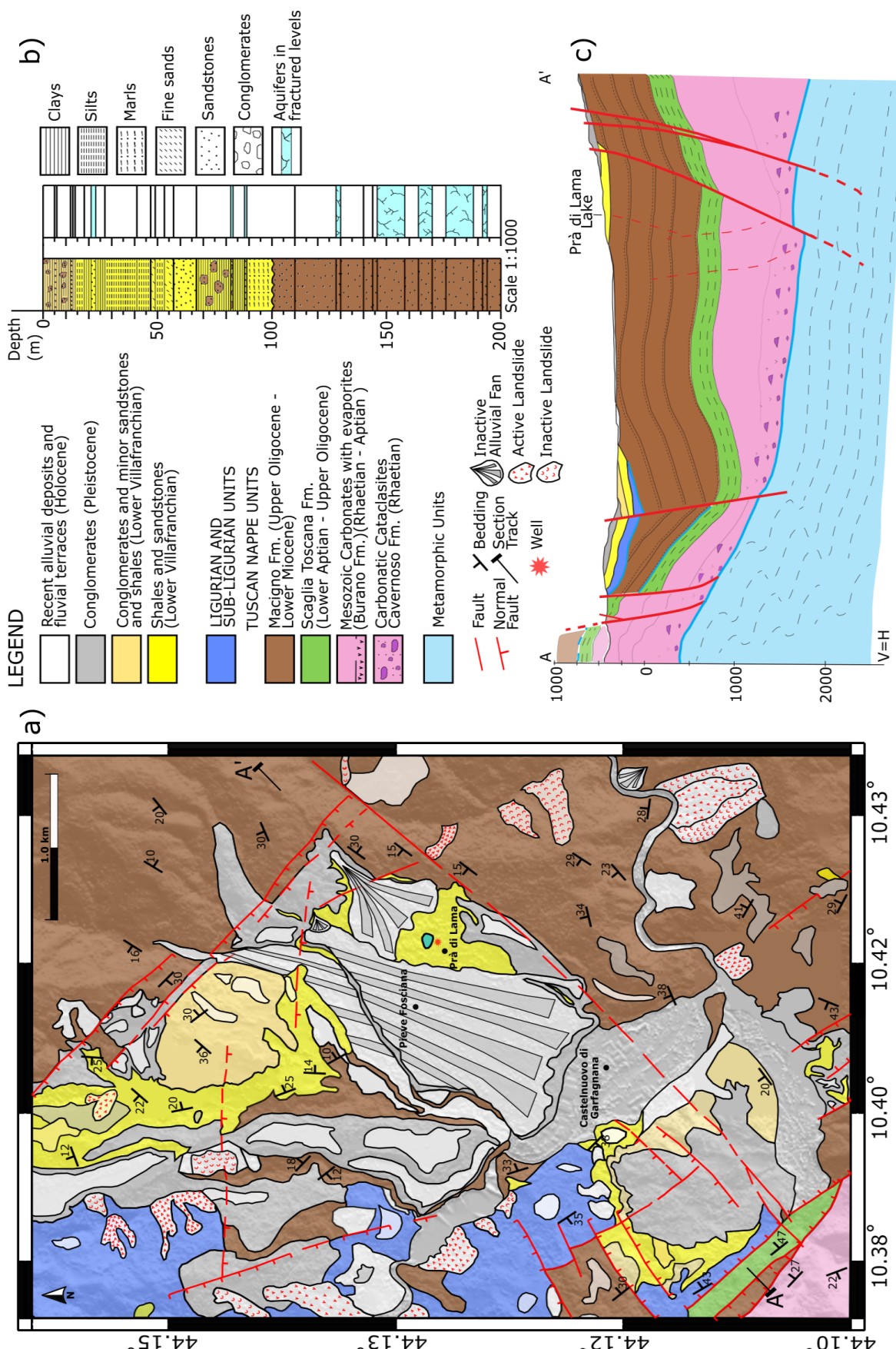

**Figure 2 – Geological setting of the study area. a)** Geological, structural and geomorphological map of the area nearby Prà di Lama
showing the main tectonic and lithostratigraphic units. **b)** Schematic sedimentary sequence of the Villafranchian deposits obtained
from the well drilled at Prà di Lama (*Modified from Chetoni 1995*). **c)** Stratigraphic cross-section across the Garfagnana graben.

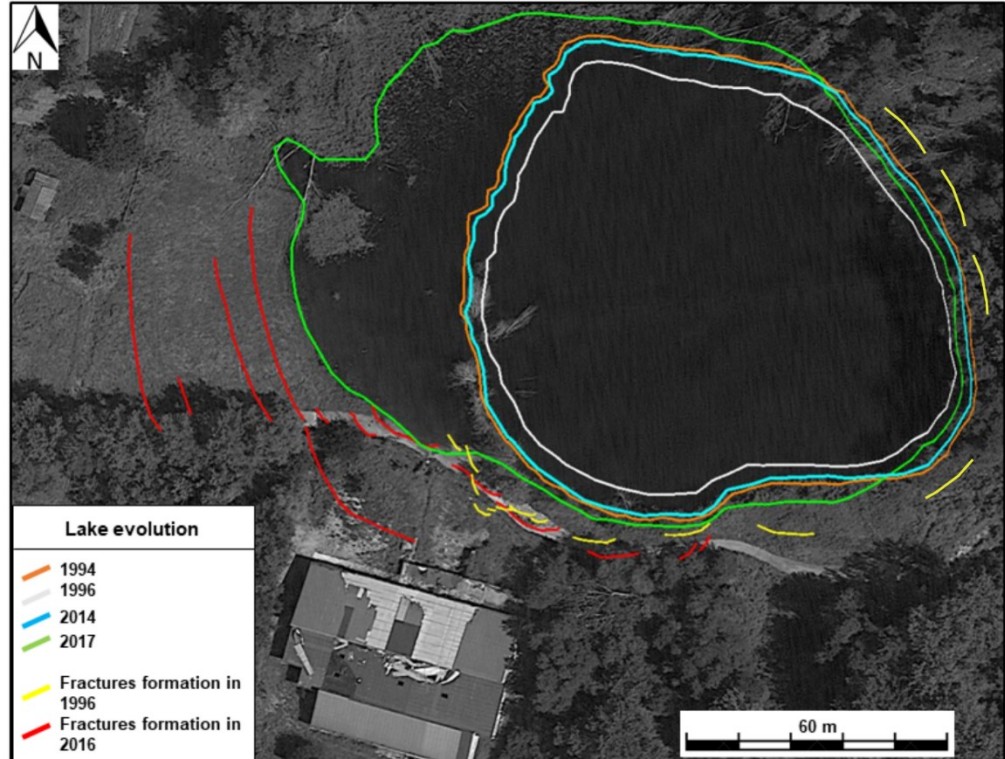

*Figure 3 – Evolution of the Prà di Lama lake between 1994 and 2017*. Lake shores variation have been retrieved from the analysis of
Landsat image

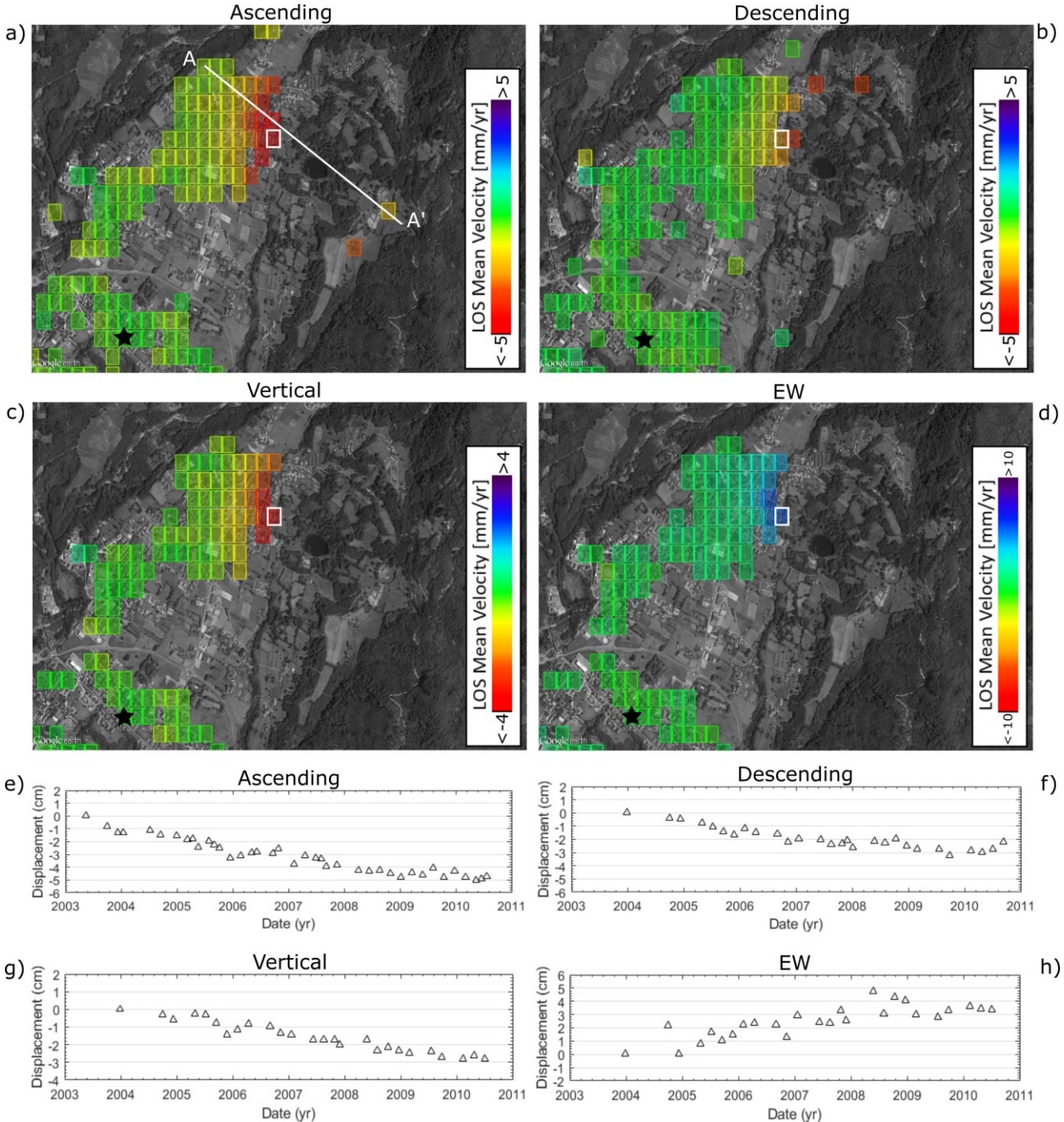



***Figure 4 – a, b)*** Maps of average surface velocity and its vertical **(c)** and East-West **(d)** components obtained from ENVISAT SAR images
acquired between 2003 and 2010. Negative values indicate range increase. The white line in panel a) marks the cross-section shown
in figure 4. The black star is the point used as reference for the InSAR-SBAS processing. **e, f, g, h)** Time-series of incremental
deformation extracted from the pixel bounded with the white rectangle.

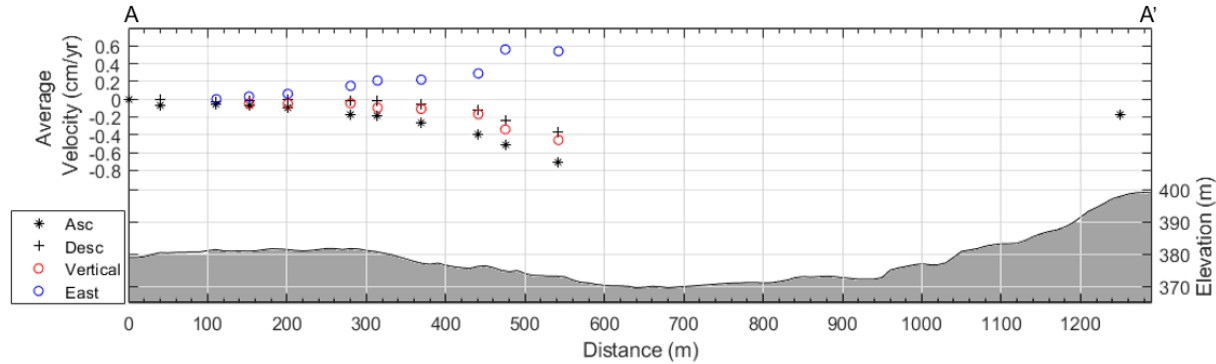


**Figure 5** - *Cross-section of topography and InSAR velocities along the A-A' profile as shown in figure 3a.*


*Figure 6 – Seismicity features of an area 10 km in radius around the Prà di Lama lake*. Cumulative seismic moment released in the
area **(a)** and histograms of the number of earthquakes per month. Three different classes of magnitude have been created: Ml < 2.0
**(b)**, 2.0 < Ml < 3.0 **(c)** and Ml > 3.0 **(d)**. The dataset covers the period between 1986 and 2017. The red transparent bars indicate the
two events of unrest of 1996 and 2016.

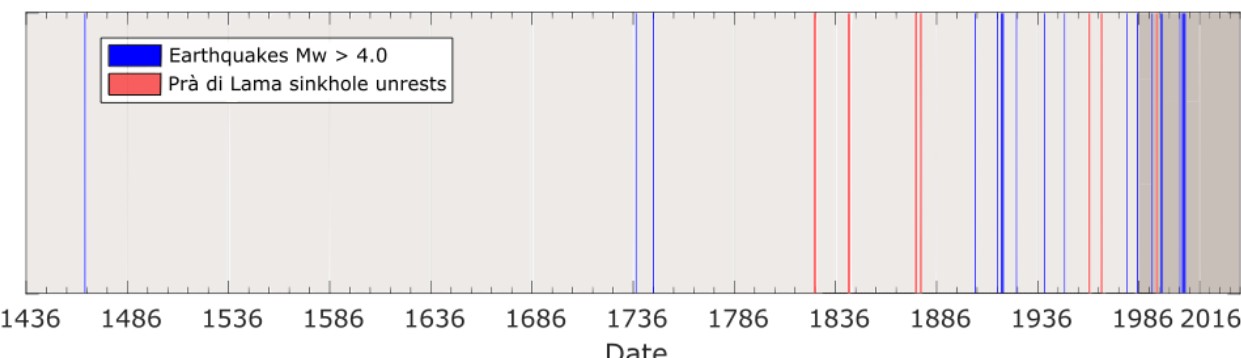

*Figure 7 - Seismicity features of an area 3 km in radius around the Prà di Lama lake*. Plot of the cumulative seismic moment released
in the area **(a)** and histograms showing the number of earthquakes occurred each month. Two different classes of Magnitude have
been created: Ml < 2.0 **(b)**, 2.0 < Ml < 3.0 **(c)**. No events of Ml > 3.0 occurred in the area between 1986 and 2017. The red transparent
bars indicate the two events of unrest of 1996 and 2016.


*Figure 8 –* Comparison between the earthquakes (blue lines) in the Garfagnana area (*INGV Catalogo Paramentrico dei Terremoti*
*Italiani CPTI15, Rovida et al., 2016*), and events of unrest at the Prà di Lama sinkhole (red lines).


| Year | Brief description of the event |
|---|---|
| 991 | Seasonal pool fed by springs |
| 1828 | Bursts of the springs water flow. Uprising of muddy waters and clays (*Raffaelli, 1869; De Stefani, 1879*) |
| 1843 | Bursts of the springs water flow. Uprising of muddy waters and clays (*Raffaelli, 1869; De Stefani, 1879*) |
| 1876 | Subsidence and fracturing (*De Stefani, 1879*) |
| 1877 | Subsidence and fracturing (*De Stefani, 1879*) |
| 1962 | Bursts of the spring water flow. Uprising of muddy waters and clays (*Giovannetti, 1975*) |
| 1969 | Abrupt falling of the water level and fracturing along the shores. The lake almost disappeared (*Giovannetti, 1975*) |
| 1985 | Arising of muddy waters in a well |
| 1996 | Abrupt fall of the water level and fracturing along the shores |
| 2016-2017 | Subsidence and fracturing |

*Table 1 – Description of the activity at Prà di Lama lake*