# Peer review of "GROWTH OF A SINKHOLE IN A SEISMIC ZONE OF THE NORTHERN APENNINES (ITALY)"

_Natural Hazards and Earth System Sciences, 2018_

## Referee Comment (RC1) · Anonymous Referee #1 · 6 Apr 2018

1). general comments

This paper presents interesting reflexions. However, there are fundamental knowledges that are incompletely presented and others that are missing. Hence, in its present state, the document should be considered as a first iteration regarding a topic that deserve to be analyzed. The methodology should be strengthened and the workflow combining different sources of data clarified. I suggest major revision.

2). specific comments

About the strategy: the comparison of four independent datasets is a good strategy to deduce a model. However:

- the few number of independent sources (SBAS, historical data, field survey, and seismic analysis);

- their variable quality levels (field observations not enough extended);

- their limited nature (there is no geomorphological map, no structural map, no trenching, no SBAS field validations, no sub-surface geophysics, no boreholes);

- their very partial temporal and spatial overlap (most information are concentrated in the last two decades);

does not allow a clear understanding of the sinkhole formation.

The idea of seismic creep seems to me not supported by a robust analysis performed at local and regional scales. The sub-surface geophysical facet is missing and therefore it is very difficult to be convinced with this explanation. Much deeper investigations are still needed.

The authors are performing some comparisons with the Dead Sea sinkholes. In Israel, lot of geophysical studies have been performed in the last 15 years to create a robust model combining geomorphological mapping, structural inputs, Insar ground deformations and shallow geophysical study results (e.g. Ezersky et al.). In this paper, most of the data are not sufficient to quantify/to observe a possible link between seismic creep and the dynamic of the collapsed area. Aware of the literature regarding the Dead Sea sinkholes, I would like to point out the attention of the authors on a circular depression located in the Jordanian Dead Sea zone and named "Birkat El Haj". It is described as a salt collapse structure. A priori, it seems to me that a comparison in term of genesis could be established.

The goal of using different data sources is to find independent observations leading to the same conclusions. In that framework, opposite conclusions are also interesting because they highlight knowledge gaps in the overall strategy. Having said that, here are some remarks on the data sources:

- the historical records is a too limited set of observations. They are informative but

could become much more relevant if they were complemented by trenching and dating as it is done in paleo-seismology in combination with historical data;

- the authors described the depression as a circular feature. However, the analysis of the contours indicates that the depression is more elliptical than circular. The lowest elevations (lake) are not located in the center of the ellipse but rather in the SW side. This asymmetry and the cracks mapped during the field survey suggest a gradual migration SW wards from the original collapse. Is this SW-NE direction important with regard to the structural data in the region? If validated, this interpretation would means that trenches could be excavated in the NE part of the depression to potentially reveal former shorelines of the lake;

- the SBAS analysis presents interesting results but the reference point is not indicated. Besides, what is the stability of the reference point chosen?

- SBAS deformation pattern suggests that the subsidence area is much wider than the actual depression revealed by contour lines. SBAS coherence threshold 0.8 is much too high and a map with coherence level at 0.4-0.5 should be drawn to try to display the whole deformation pattern. Of course there will be much more noise but this is the conditions to get the maximum from the images;

- SBAS points selected with coherence at 0.8 level indicated important ground movements that should have created series of fissures and fractures in the buildings of the nearby village. The collection of those pieces of evidence is necessary to validate the SBAS observations. Further more, those evidences should be linked to the structural context of the depression;

- structural map is not presented while this source of data is interesting to link the genesis of the depression with a possible seismic creep;

- geomorphological maps, at local and regional scales, should be drawn in order to confirm that this depression is really a singularity in the landscape. There is no evi-

dence anywhere that this depression is an isolate case or that similar phenomena can be observe elsewhere in the region. It is really important to clarify the status of this depression because if it is an isolated case, then, it can be considered a very interesting indicator regarding the tectonic activity in the region;

- the stratigraphy is very poorly described and the thickness of the different layers below the depression is incomplete. A carbonate layer is mentioned in the text (Tuscan Nappe Unit) but not its depth while this layer is a good candidate to be the siege of dissolution phenomena leading to ground subsidence at the surface.

3). technical corrections": typing errors, etc.

n/a at this stage.

4). Q&A

- Does the paper address relevant scientific and/or technical questions within the scope of NHESS?

yes

- Does the paper present new data and/or novel concepts, ideas, tools, methods or results?

yes

- Are these up to international standards?

yes

- Are the scientific methods and assumptions valid and outlined clearly?

no

- Are the results sufficient to support the interpretations and the conclusions?

no

- Does the author reach substantial conclusions?

yes

- Is the description of the data used, the methods used, the experiments and calculations made, and the results obtained sufficiently complete and accurate to allow their reproduction by fellow scientists (traceability of results)?

no

- Does the title clearly and unambiguously reflect the contents of the paper?

no

- Does the abstract provide a concise, complete and unambiguous summary of the work done and the results obtained?

no

- Are the title and the abstract pertinent, and easy to understand to a wide and diversified audience?

no

- Are mathematical formulae, symbols, abbreviations and units correctly defined and used? If the formulae, symbols or abbreviations are numerous, are there tables or appendixes listing them?

n/a

- Is the size, quality and readability of each figure adequate to the type and quantity of data presented?

no

- Does the author give proper credit to previous and/or related work, and does he/she indicate clearly his/her own contribution?

yes

- Are the number and quality of the references appropriate?

yes

- Are the references accessible by fellow scientists?

yes

- Is the overall presentation well structured, clear and easy to understand by a wide and general audience?

yes

- Is the length of the paper adequate, too long or too short?

yes

- Is there any part of the paper (title, abstract, main text, formulae, symbols, figures and their captions, tables, list of references, appendixes) that needs to be clarified, reduced, added, combined, or eliminated?

figures 1 & 2 should be redrawn

- Is the technical language precise and understandable by fellow scientists?

yes

- Is the English language of good quality, fluent, simple and easy to read and understand by a wide and diversified audience?

yes

- Is the amount and quality of supplementary material (if any) appropriate?

n/a

2018-75, 2018.

---

## Referee Comment (RC2) · Anonymous Referee #2 · 11 Apr 2018

1. General comments: The authors present an interesting piece of work with interpretations on the activity of one sinkhole in a seismically active zone. Essentially, the work proposes the following conclusions/interpretations: (1) The dynamics of the analysed sinkhole, characterised by progressive subsidence, punctuated by events of more rapid displacement and ground fissuring (1996, 2016), are attributed to creeping faults in the area that induce fracturing, permeability increase and enhanced dissolution. (2) Based on DInSAR data, ground deformation affects a large area around the sinkhole lake with horizontal displacement rates as high as the vertical ones. However, I consider that such conclusions/interpretations are not properly justified, and authors should consider and discuss other alternative interpretations. Concerning point (1), authors should also consider other potential controlling factors such as precipitation

and groundwater level changes. Moreover, the available data does not seem to be sufficient to rule out the role of major morphogenetic earthquakes on sinkhole triggering. Authors should review the existing literature that document the formation of coseismic sinkholes in Italy. Regarding point (2), authors should consider the option that ground displacement with significant horizontal component on the NW margin of the sinkhole could be related to a landslide, favoured by debuttressing-undermining at the foot of the slope due to sinkhole subsidence.

2. Specific comments:

2.1. The paper lacks essential data on the geomorphic context, including a detailed map. The latter may show the presence of landslides or other sinkholes in the area. A thorough geomorphological analysis is needed to identify the active processes in the study area and distinguish their relative importance in the sinkhole deformation dynamic. Such as: detailed mapping, trenching combined with geochronological data (to study the geological record and increase the temporal registry), and geophysics.

2.2. I believe the sinkhole definition used (lines 29-30) is inadequate since not all the sinkholes form due to cavity collapse. There are other genetic processes. The authors should clearly indicate the type of sinkhole they are investigating, explaining the subsidence mechanisms in relationship with the local stratigraphy. I consider that revising this paper: Parise, M., Closson, D., Gutiérrez, F. et al. Environ Earth Sci (2015) 74: 7823. https://doi.org/10.1007/s12665-015-4647-5; could help. The cover is underlain by flysch. Do you have deep-seated caprock collapse sinkholes?

2.3. The authors conclude that "a source mechanism for the sinkhole formation and growth is seismic creep in the active fault zone underneath the sinkhole". Although this hypothesis looks innovative, it is not well supported by the presented data. The casual relationship between creep tectonic deformation and sinkhole activity remains as an unproved hypothesis. I encourage the authors to add sub-subface geophysical and structural data to test their hypothesis.

3. Technical corrections: I have identified two technical issues.

3.1. (lines 187 and 191) Change 4a for 5a.

3.2. Add a legend to the figures 5 and 6 to help to identify the meaning of the red and yellow lines.

4. Other aspects to take into account:

4.1. Does the paper address relevant scientific and/or technical questions within the scope of NHESS? Yes

4.2. Does the paper present new data and/or novel concepts, ideas, tools, methods or results? Yes

4.3. Are these up to international standards? Yes

4.4. Are the scientific methods and assumptions valid and outlined clearly? No

4.5. Are the results sufficient to support the interpretations and the conclusions? No

4.6. Does the author reach substantial conclusions? Yes

4.7. Is the description of the data used, the methods used, the experiments and calculations made, and the results obtained sufficiently complete and accurate to allow their reproduction by fellow scientists (traceability of results)? No

4.8. Does the title clearly and unambiguously reflect the contents of the paper? Yes

4.9. Does the abstract provide a concise, complete and unambiguous summary of the work done and the results obtained? No

4.10. Are the title and the abstract pertinent, and easy to understand to a wide and diversified audience? Yes

4.11. Are mathematical formulae, symbols, abbreviations and units correctly defined and used? If the formulae, symbols or abbreviations are numerous, are there tables or

appendixes listing them? Yes

4.12. Is the size, quality and readability of each figure adequate to the type and quantity of data presented? No, legend missing in figures 5 and 6.

4.13. Does the author give proper credit to previous and/or related work, and does he/she indicate clearly his/her own contribution? Yes

4.14. Are the number and quality of the references appropriate? Yes

4.15. Are the references accessible by fellow scientists? Yes

4.16. Is the overall presentation well structured, clear and easy to understand by a wide and general audience? Yes

4.17. Is the length of the paper adequate, too long or too short? Is adequate

4.18. Is there any part of the paper (title, abstract, main text, formulae, symbols, figures and their captions, tables, list of references, appendixes) that needs to be clarified, reduced, added, combined, or eliminated? A geomorphological/geological map would help to understand the setting and active processes.

4.19. Is the technical language precise and understandable by fellow scientists? Yes

4.20. Is the English language of good quality, fluent, simple and easy to read and understand by a wide and diversified audience? Yes

4.21. Is the amount and quality of supplementary material (if any) appropriate? -not applicable-

---

## Short Comment (SC1) · 2 May 2018

Figure 1 - a) result of the first stage of the processing setting the reference point at Massa city (Black star). Please note the stability south of Pieve Fosciana in the white rectangle. b) Close up on the Pieve Fosciana area showing the final reference point (white star).

Figure 2 - a, b) Results of the processing with coherence threshold of 0.8. c, d) Results of the test with coherence threshold of 0.4.

Figure 3 - Historical record of the events occurred at Prà di Lama (red lines) and earthquakes occurred in the Garfagnana graben (blue lines). The historical seismicity record has been obtained from the CPTI15 catalogue. The light grey area represents the historical seismicity while the dark grey area corresponds to the instrumental seismicity. Please note the lack of correspondence between earthquakes and events of unrest.

[Figure]

---

## Author Comment (AC1) · 2 May 2018

We thank both anonymous referees for their constructive comments. We have addressed all of them in the following point-to-point rebuttal letter and we will incorporate the changes in a revised manuscript. Specifically, we have strengthened the case for the presence of an active sinkhole in Pieve Fosciana by adding a geomorphological and structural map, a geological cross-section and a stratigraphic log from borehole data. Furthermore, we expanded the seismicity analysis and added a figure clearly showing lack of correlation between distant earthquakes and sinkhole unrest. We also included a description of a recent geochemical study showing that fluids at Prà di Lama migrate along faults from a carbonatic reservoir at 2 km depth. We believe that the additional evidences show to greater confidence that the Prà di Lama lake is a sinkhole

whose formation and growth is likely controlled by the active faults system. Finally, we developed a direct comparison of our study to other areas where faults creep to put our results in the broader tectonic context.

As comments by the reviewers have some common themes, we have sorted each reviewer's comments and grouped those that have common themes. We also put our responses to the reviewers' comments in italics. In detail:

**Reviewer 1. - the few number of independent sources (SBAS, historical data, field survey, and seismic analysis);**

- their variable quality levels (field observations not enough extended);

- their limited nature (there is no geomorphological map, no structural map, no trenching, no SBAS field validations, no sub-surface geophysics, no boreholes);

- structural map is not presented while this source of data is interesting to link the genesis of the depression with a possible seismic creep;

- geomorphological maps, at local and regional scales, should be drawn in order to confirm that this depression is really a singularity in the landscape. There is no evidence anywhere that this depression is an isolate case or that similar phenomena can be observe elsewhere in the region. It is really important to clarify the status of this depression because if it is an isolated case, then, it can be considered a very interesting indicator regarding the tectonic activity in the region;

- the stratigraphy is very poorly described and the thickness of the different layers below the depression is incomplete. A carbonate layer is mentioned in the text (Tuscan Nappe Unit) but not its depth while this layer is a good candidate to be the siege of dissolution phenomena leading to ground subsidence at the surface.

**Reviewer 2, specific comment 2.1:**

The paper lacks essential data on the geomorphic context, including a detailed map.

The latter may show the presence of landslides or other sinkholes in the area. A thorough geomorphological analysis is needed to identify the active processes in the study area and distinguish their relative importance in the sinkhole deformation dynamic. Such as: detailed mapping, trenching combined with geochronological data (to study the geological record and increase the temporal registry), and geophysics.

Response: We agree with the reviewers that a better description of the stratigraphy, geology, tectonic structures, and geomorphology of the Pieve Fosciana area is needed to improve the paper, providing more compelling evidences of the presence of an active sinkhole and its relationship to tectonics. Therefore, we will add a geomorphological and structural map as well as a geological cross-section of the study area. This will show the stratigraphy of the Prà di Lama sinkhole and the presence, at $\sim$ 2 km depth, of carbonatic and evaporitic formations. Furthermore, a detailed geomorphological map will be added to show the presence of the Prà di Lama sinkhole and the lack of landslides in the area affected by ground motion as identified by InSAR. The Prà di Lama sinkhole is an isolated feature in the region being the only mapped sinkhole in the entire Garfagnana graben (Caramanna et al., 2008); the closest sinkholes are near the Tuscany coast. We will add to figure 1 the locations of the sinkholes to clearly show the Prà di Lama site is an isolated case. We will also expand on the field observations by adding photos of fractures and cracks around the lake that further document the deformation activity at Prà di Lama. Observations from a 200 m-deep borehole will also be added, showing the detailed stratigraphy. A recent study from Gherardi and Pierotti (2018), expanding on previous research (Baldacci et al., 2007), uses geochemical analyses of the Prà di Lama spring waters and concludes that the high salinity ($\sim$5.9g/kgw) and temperature ($\sim$57 °C) are explained by hydrothermal circulation between 1.3 and 2 km depth in an evaporitic-carbonate reservoir. The results from this study are in agreement with the presence of a deep sinkhole at Prà di Lama, and the evaporitic-carbonate reservoir likely corresponds to the anhydrite of the Burano Fm. and the calcareous-dolomitic breccia of the Calcare Cavernoso Fm., that will be shown by our geologic cross-section. More importantly, the un-mixing of deep waters in shallow aquifers is interpreted by Gherardi and Pierotti (2018) as an evidence of the rapid upwelling of deep waters along the main tectonic structures (Baldacci et al., 2007). The conclusions reached will be added to the manuscript in the discussion section. Our revised dataset will include observations from completely independent methods spanning surface deformation (InSAR), structural geology, seismicity, geomorphology, borehole stratigraphy and a century-long historical record of sinkhole activity. We also added a comparison of our results to independent geochemistry results. Several independent lines of evidence that support our conclusions to greater confidence are now provided. Data from sub-surface geophysics and trenching in Pieve Fosciana are not available and their high cost prevents us to obtain these datasets at present. While these data would provide a more detailed view of the source geometry and fault activity, we believe that the main conclusions reached by our study are supported. We hope that our study will raise the scientific interest in the area and that studies of sub-surface geophysics and trenching will be carried out in the future.

**Reviewer 1. - Partial temporal and spatial overlap. (most information are concentrated in the last two decades);**

- The historical record is a too limited set of observations. They are informative but could become much more relevant if they were complemented by trenching and dating as it is done in paleo-seismology in combination with historical data does not allow a clear understanding of the sinkhole formation. The idea of seismic creep seems to me not supported by a robust analysis performed at local and regional scales. The sub-surface geophysical facet is missing and therefore it is very difficult to be convinced with this explanation. Much deeper investigations are still needed.

Response: The first information about the Prà di Lama lake date back to the 991 A.D., when it was reported that a depression filled by a lake formed from a series of previously isolated springs. We will add this explanation to the manuscript to clarify that the historical record allows us to define the time-scale of sinkhole formation. Furthermore, the historical record shows well the episodic behaviour of the sinkhole; this will also be

clarified in the manuscript. We agree with the reviewer that the historical record is limited and therefore we complemented it with InSAR, seismicity and structural geology data that have now been expanded to include geomorphology, borehole stratigraphy and a comparison to a recently published geochemistry study. See also our previous response. Regarding the seismicity analysis, we used a reputed catalogue, the Italian national catalogue of seismicity recorded by the Istituto Nazionale di Geofisica e Vulcanologia (INGV) and available online (http://cnt.rm.ingv.it/search). We already analysed the seismic moment release and magnitude contents both in the broader tectonic region and at the local scale. The complete dataset containing all the low magnitude events (Ml < 2.0) dates back to the 1986. Other historical catalogue exists, but earthquakes locations are not accurate, nor are the magnitudes. This makes it difficult to attribute an earthquake to one of the faults nearby the lake. Thus, we decided to limit our analysis to seismic data from 1986 and we will clarify this in the manuscript. To strengthen our tectonic interpretation, we expanded the section on fault creep providing examples where a comparable fault activity has been suggested. We will show examples from Taiwan (Rau et al., 2007; Chen et al., 2008; Harris, 2017) and Parkfield (California) (Nadeau et al., 1995, Harris, 2017). InSAR is a fairly recent technique and we do not have data past the last few decades thus we have complemented our observations of recent subsidence to structural geology providing stronger evidences of the longer-term tectonic activity in the area. We will add a more detailed structural geology map showing the faults in the area.

Comment: The authors are performing some comparisons with the Dead Sea sinkholes. In Israel, lot of geophysical studies have been performed in the last 15 years to create a robust model combining geomorphological mapping, structural inputs, InSAR ground deformations and shallow geophysical study results (e.g. Ezersky et al.). In this paper, most of the data are not sufficient to quantify/to observe a possible link between seismic creep and the dynamic of the collapsed area. Aware of the literature regarding the Dead Sea sinkholes, I would like to point out the attention of the authors on a circular depression located in the Jordanian Dead Sea zone and named "Birkat El Haj".

It is described as a salt collapse structure. A priori, it seems to me that a comparison in term of genesis could be established.

Response: We thank the referee for pointing us to this relevant paper. We will add to our discussion section the example of the Birkat El Haj sinkhole where a relationship with tectonic structures has been inferred. The similarities between strike-slip tectonics in Birkat El Haj and Prà di Lama is particularly relevant.

Comment: the authors described the depression as a circular feature. However, the analysis of the contours indicates that the depression is more elliptical than circular. The lowest elevations (lake) are not located in the centre of the ellipse but rather in the SW side. This asymmetry and the cracks mapped during the field survey suggest a gradual migration SW wards from the original collapse. Is this SW-NE direction important with regard to the structural data in the region? If validated, this interpretation would means that trenches could be excavated in the NE part of the depression to potentially reveal former shorelines of the lake

Response: The temporal reconstruction shows that the lake had a quasi-circular shape between 1994 and 2014. The events of 1996, in fact, consisted in a lake-level fluctuation accompanied by slumping of the shores and cracks formation (yellow lines in Figure 2). No significant changes in lake's shape or dimension occurred in 1996. The elliptical shape results from the last event of 2016. During that event fracture formed in the SW as a consequence of subsidence. The main active strike-slip fault is also oriented SW, suggesting a tectonic control on the deformation and in agreement with our interpretation of tectonic-induced sinkhole. This explanation will be added to the manuscript. Figure 2 will also be modified as requested and a structural map will be added in our manuscript together with photos that better document the most recent phenomena.

Comment: The SBAS analysis presents interesting results but the reference point is not indicated. Besides, what is the stability of the reference point chosen?

Response: Although the location of the reference point was indicated in our original manuscript as a black star in figure 3, we acknowledge its visibility should be improved and we have modified the figures as below. In any case, the reference point was initially set in the city of Massa because GPS measurements from Bennett et al. (2012) show that the surface velocities there are < 1mm/yr, therefore, Massa can be considered stable. Assuming Massa as reference point, the retrieved LOS deformation maps revealed the deformation pattern around the Pieve Fosciana (Fig. 1a). We then selected a reference point outside the deformation pattern but close to Pieve Fosciana town (white star in figure 1b). The reason of this change (that can be done in post processing and does not affect the result accuracy) is due to the fact that the new reference point is closer to our deformation signal than Massa, allowing us to reduce the impact of the atmospheric artefacts in the LOS displacement time series. The tropospheric artefacts are spatially correlated and thus can be considered almost identical in areas close to the reference point. Therefore, by using a reference point close to the deformation signal, the impact of tropospheric disturbances can be minimized. This procedure in summary implies that the quality of the final measurements is improved in the area under study.

Comment: - SBAS deformation pattern suggests that the subsidence area is much wider than the actual depression revealed by contour lines. SBAS coherence threshold 0.8 is much too high and a map with coherence level at 0.4-0.5 should be drawn to try to display the whole deformation pattern. Of course, there will be much more noise but this is the conditions to get the maximum from the images.

Response We chose a coherence threshold of 0.8 in order to guarantee that only reliable pixels are analysed and interpreted here. The 0.8 value is referred to the threshold used for the phase unwrapping step to identify the pixels that will be unwrapped by the Extended Minimum Cost Flow (EMCF) algorithm. By setting high values of this parameter the pixels in input to the EMCF algorithm are affected by less noise thus increasing the quality of the phase unwrapping step itself. This approach has shown

to be also effective in areas affected by low coherence values (Cignetti et al. 2016). In any case, according to the Reviewer's request, we reprocessed the data using a coherence threshold of 0.4. The new results are displayed below (Fig. 2) and show that only a few more pixels are gained north of the sinkhole as compared to our original results. We conclude that by decreasing the coherence threshold we cannot retrieve the whole deformation pattern. This is likely due to the fact the area is highly vegetated. Comment - SBAS points selected with coherence at 0.8 level indicated important ground movements that should have created series of fissures and fractures in the buildings of the nearby village. The collection of those pieces of evidence is necessary to validate the SBAS observations. Furthermore, those evidences should be linked to the structural context of the depression. Response:

There have been no reports of fractures or fissures in the buildings of the village, this is likely because the subsidence pattern is relatively broad compared to the size of a building thus there is no significant strain applied to the buildings. Structural damages are the consequence of high strain rates applied to individual buildings (Arangio et al., 2013). The only presence of fractures occurred in an abandoned building in the immediate vicinity of the lake and we will add photos of these in the revised manuscript. However, InSAR is not coherent in this area and a direct comparison between the deformation field and the building damage cannot be derived.

**Reviewer 2**

General Comments: The authors present an interesting piece of work with interpretations on the activity of one sinkhole in a seismically active zone. Essentially, the work proposes the following conclusions/interpretations: (1) The dynamics of the analysed sinkhole, characterised by progressive subsidence, punctuated by events of more rapid displacement and ground fissuring (1996, 2016), are attributed to creeping faults in the area that induce fracturing, permeability increase and enhanced dissolution. (2) Based on DInSAR data, ground deformation affects a large area around the sinkhole lake with horizontal displacement rates as high as the vertical ones. However, I

consider that such conclusions/interpretations are not properly justified, and authors should consider and discuss other alternative interpretations. Concerning point (1), authors should also consider other potential controlling factors such as precipitation and groundwater level changes. Moreover, the available data does not seem to be sufficient to rule out the role of major morphogenetic earthquakes on sinkhole triggering. Authors should review the existing literature that document the formation of coseismic sinkholes in Italy. Regarding point (2), authors should consider the option that ground displacement with significant horizontal component on the NW margin of the sinkhole could be related to a landslide, favoured by debuttressing-undermining at the foot of the slope due to sinkhole subsidence.

Response We are glad that the reviewer finds our results interesting. We agree that seismic creep is one interpretation but other possible source mechanisms should be addressed. We will include a geomorphologic map of the Pieve Fosciana area showing that no landslide has been identified in the actively deforming area. Furthermore, the low topographic slope rules out the presence of an active landslide. On the other hand, a recent geochemical study (Gherardi and Pierrotti, 2018) shows that waters at Prà di Lama raise from a deep aquifer ($\sim$2000 m) along a fractures system. This is in agreement with the presence of a fault and a deep pipe sinkhole. The horizontal eastward motion derived by InSAR is in agreement with contraction toward the centre of the sinkhole as a result of subsidence. We will add this discussion to the paper.

Precipitations can influence the groundwater level and thus ground motions but these patterns have a seasonal trend rather than continuous subsidence over a timespan of several years, as shown by our InSAR analysis. A long-term subsidence could potentially be caused by over-exploitation of an aquifer but no water is pumped from the aquifers in the deforming area. We will add this explanation to the manuscript. Furthermore, the broad subsidence pattern observed a Pieve Fosciana (see figure 3 and 4) indicates a deep source, likely the 2 km depth carbonatic-evaporite formation. The hypothesis of seismic creep along an active fault remains our favourite interpretation because this mechanism can explain the variety of observations, ranging from surface subsidence as seen by InSAR, lake level fluctuations documented in the historical record, mapped faults from structural geology and upward fluid migration from geochemistry.

Although the relationship between active faults, creep and surface features, like sinkholes, is a relatively new research topic, it is well established that faults creep both seismically and aseismically (e.g. Linde et al. 1996; Wei et al. 2013). In particular, seismic creep has been reported along different active faults (i.e. Linde et al. 1996, Nadeau et al., 1995; Rau et al., 2007; Chen et al., 2008; Harris, 2017). Relationships between creeping faults and fluid migration causing enhanced permeability are also widely reported in literature (i.e. Wei et al., 2009; Scholz, 1998; Yarushina et al., 2017; Sibson, 1996; Micklethwaite et al., 2010). These observations justify the hypothesis of seismic creep at Prà di Lama because of the presence of an active faults, evidences of deep fluid migration and a mapped sinkhole. We will add this explanation to the manuscript together with an expanded section detailing the above-mentioned examples of seismic creep.

To strengthen our seismicity analysis and clarify whether a connection between major tectonic earthquakes and sinkhole unrest exists, we analysed both the historical and instrumented seismic catalogues (INGV Catalogo Parametrico dei Terremoti Italiani, CPTI15). We now include a new figure (Fig.3) showing the occurrence of major earthquakes, with magnitude > 4.0 up to 20 km distant from Pieve Fosciana and the recorded events of unrest at the sinkhole. The figure shows that there is no clear connection between occurrence of large distant earthquakes and events of sinkhole unrest, therefore the mechanisms responsible for the Prà di Lama sinkhole formation should be attributed to local processes.

Comment 2.2: I believe the sinkhole definition used (lines 29-30) is inadequate since not all the sinkholes form due to cavity collapse. There are other genetic processes. The authors should clearly indicate the type of sinkhole they are investigating, explaining the subsidence mechanisms in relationship with the local stratigraphy. I consider that revising this paper: Parise, M., Closson, D., Gutiérrez, F. et al. Environ Earth Sci (2015) 74: 7823. https://doi.org/10.1007/s12665-015-4647-5; could help. The cover is underlain by flysch. Do you have deep-seated caprock collapse sinkholes?

Response: We will modify the sentence "Sinkholes are quasi-circular depressions in the ground surface that form due to the breakdown of subterranean cavities" to "Sinkholes are closed depressions typically associated to karst environments", in accordance with Ford and Williams (2007) and Gutierrez et al. (2014). We will than add a brief description of the main genetic processes, as suggested by the reviewer, and clarify that we study a sinkhole classified as Deep Piping Sinkhole, according to Caramanna et al. (2008). We will explain the subsidence mechanisms in relationship with the local stratigraphy by clarifying that collapse of deep cavities caused by fluid circulation occurs in carbonatic-evaporitic formations located at 1.3-2 km depth and covered by a thick non-carbonatic sequence. The Prà di Lama sinkhole can be defined as a 'deep-seated caprock collapse sinkhole' according to Gutierrez et al. (2008) and Parise et al. (2015) and we will also add this definition to the manuscript for clarity.

Comment 2.3: The authors conclude that "a source mechanism for the sinkhole formation and growth is seismic creep in the active fault zone underneath the sinkhole". Although this hypothesis looks innovative, it is not well supported by the presented data. The casual relationship between creep tectonic deformation and sinkhole activity remains as an unproved hypothesis. I encourage the authors to add sub-subface geophysical and structural data to test their hypothesis.

Response: We agree that the explanation of a tectonic-induced sinkhole is new and we will provide a new structural geology map as well as a geology cross-section showing that faults geometries are consistent with a structural control on the sinkhole. We will also add a discussion section including examples of active faults characterized by seismic creep analogous to our case. In particular we will present the examples of Taiwan (Rau et al., 2007; Chen et al., 2008; Harris, 2017) and Parkfield (California)

(Nadeau et al., 1995, Harris, 2017).

References:

- Arangio, S., Calò, F., Di Mauro, M., Bonano, M., Marsella, M., and Manunta, M. An application of the SBAS-DInSAR technique for the assessment of structural damage in the city of Rome. Struct. Infrastruct. Eng. Maintenance, Manag. Life-Cycle Des. Perform. 1–15 DOI: 10.1080/15732479.2013.833949 (2013)

- Baldacci, F., Botti, F., Cioni, R., Molli, G., Pierotti, L., Scozzari, A., Vaselli, L., 2007. Geological-structural and hydrogeochemical studies to identify sismically active structures: case history from the Equi Terme-Monzone hydrothermal system (Northern Apennine – Italy). Geoitalia, 6th Italian Forum of Earth Sciences. Rimini, 2007.

- Chen, K.H., Nadeau, R.M., Rau, R. Characteristic repeating earthquakes in an arc-continent collision boundary zone: The Chihshang fault of eastern Taiwan. Earth and Planetary Science Letters. DOI: 10.1016/j.epsl.2008.09.021 (2008)

- Cignetti, M., Manconi, A., Manunta, M., Giordan, D., De Luca, C., Allasia, P., Ardizzone, F. Taking Advantage of the ESA G-POD Service to Study Ground Deformation Processes in High Mountain Areas: A Valle d'Aosta Case Study, Northern Italy. Remote Sens. 8, 852. DOI: 10.3390/rs8100852 (2016) Ford, D.C., Williams, P. Karst Hydrogeology and Geomorphology. Wiley, Chichester (2007)

- Gherardi, F., Pierotti, L. The suitability of the Pieve Fosciana hydrothermal system (Italy) as a detection site for geochemical seismic precursors. Applied Geochemistry DOI: 10.1016/j.apgeochem.2018.03.009 (2018)

- Gutierréz, F., Guerrero, J., Lucha, P. A genetic classification of sinkholes illustrated from evaporite paleokarst exposures in Spain. Environmental Geology (53) DOI: 10.1007/s00254-007-0727-5

- Gutierréz, F., Parise, M., De Waele J., Jourde, H. A review on natural and human-induced geohazards and impacts in karst. Earth-Science Reviews,138. DOI:

10.1016/j.earscirev.2014.08.002 (2014)

- Linde, A.T., Gladwin M.T., Johnston M.J.S., Gwyther R.L. and Bilham R.G. A slow earthquake sequence on the San Andreas fault. Nature, 383. DOI: 10.1038%2F383065a0 (1996)

- Pepe, A. and Lanari, R. On the extension of the minimum cost flow algorithm for phase unwrapping of multitemporal differential SAR interferograms. IEEE Trans. Geosci. Remote Sens., 44, 9, 2374–2383 DOI: 10.1109/TGRS.2006.873207 (2006)

- Nadeau, R.M., Foxal, W., McEvilly, T.V. Clusterind and Periodic Recurrence of Microearthquakes on the San Andreas Fault at Parkfield, California. Science, 267. DOI: 10.1126/science.267.5197.503 (1995)

- Rau, R., Chen, K.H., Ching, K. Repeating earthquakes and seismic potential along the northern Longitudinal Valley fault of Taiwan. Geophysical Research Letters, 34. DOI: 10.1029/2007GL031622 (2007)

- Wei, M., Kaneko, Y., Liu, Y. and McGuire, J., Episodic fault creep events in California controlled by shallow frictional heterogeneity. Nature Geoscience, 6. DOI: 10.1038/NGEO1835 (2013)
* * *
[Figure]

**Fig. 1.** a) result of the first stage of the processing setting the reference point at Massa city (Black star). Please note the stability south of Pieve Fosciana in the white rectangle. b) Close up on the Piev

**Fig. 2.** a, b) Results of the processing with coherence threshold of 0.8. c, d) Results of the test with coherence threshold of 0.4.

[Figure]

**Fig. 3.** Historical record of the events occurred at Prà di Lama (red lines) and earthquakes occurred in the Garfagnana graben (blue lines). The historical seismicity record has been obtained from the CPTI15

---

## Author Response (AR1)

**Point-by-Point reply to the comments**

We thank the editor and both anonymous referees for their constructive comments. We have addressed all of them in the following point-to-point rebuttal letter and we incorporated the changes in a revised manuscript. Specifically, we have strengthened the case for the presence of an active sinkhole in Pieve Fosciana by adding a geomorphological and structural map, a geological cross-section and a stratigraphic log from borehole data. Furthermore, we expanded the seismicity analysis and added a figure clearly showing lack of correlation between distant earthquakes and sinkhole unrest. We also included a description of a recent geochemical study showing that fluids at Prà di Lama migrate along faults from a carbonatic reservoir at 2 km depth. We believe that the additional evidences show to greater confidence that the Prà di Lama lake is a sinkhole whose formation and growth is linked to the local active tectonics.

As comments by the reviewers have some common themes, we have sorted each reviewer's comments and grouped those that have common themes.

**# Editor**

Major revisions are required to the manuscript, in order to consider it for publication. In addition to the comments by the referees, I suggest the Authors to refer to internationally recognized classification on sinkholes, such as that proposed in Gutierrez et al. (2008, 2014) rather than referencing to single publications. This well help the reader to have a better understanding of the processes authors are describing.

Further, Authors do not take into any account a number of papers dealing exactly with the topic of the manuscript, that is the relations between sinkhole and seismicity. I kindly invite the Authors to consider such references when preparing the revised version of the manuscript. Below Authors will find a list of suggested references:

Del Prete, S., Iovine, G., Parise, M., Santo, A., 2010b. Origin and distribution of different types of sinkholes in the plain areas of Southern Italy. Geodin. Acta 23, 113–127.

Gutiérrez, F., Guerrero, J., Lucha, P., 2008. A genetic classification of sinkholes illustrated from evaporite paleokarst exposures in Spain. Environ. Geol. 53, 993–1006.

Gutierrez F., Parise M., De Waele J. & Jourde H., 2014, A review on natural and human-induced geohazards and impacts in karst. Earth Science Reviews, vol. 138, p. 61-88, doi: 10.1016/j.earscirev.2014.08.002.

Iovine G. & Parise M., 2008, I sinkholes in Calabria. In: Nisio S. (a cura di) I fenomeni naturali di sinkhole nelle aree di pianura italiane. Memorie Descrittive della Carta Geologica d'Italia, vol. 85, p. 335-386.

Kawashima, K., Aydan, O., Aoki, T., Kishimoto, I., Konagal, K., Matsui, T., Sakuta, J., Takahashi, N., Teodori, S.-P., Yashima, A., 2010. Reconnaissance investigation on the damage of the 2009 L'Aquila, Central Italy earthquake. J. Earthq. Eng. 14, 817–841.

Parise, M., Perrone, A., Violante, C., Stewart, J.P., Simonelli, A., Guzzetti, F., 2010. Activity of the Italian National Research Council in the aftermath of the 6 April 2009 Abruzzo earthquake: the Sinizzo Lake case study. Proc. 2nd Int. Workshop "Sinkholes in the Natural and Anthropogenic Environment", Rome, pp. 623–641.

Santo, A., Del Prete, S., Di Crescenzo, G., Rotella, M., 2007. Karst processes and slope instability: some investigations in the carbonate Apennine of Campania (southern Italy). In: Parise, M., Gunn, J. (Eds.), Natural and Anthropogenic Hazards in Karst Areas: Recognition, Analysis, and Mitigation. Geological Society, London. 279, pp. 59–72.

**Response:** We modified our manuscript as suggested. We now use in our manuscript the international classification of sinkholes proposed by *Gutierrez et al. (2008, 2014)* see changes to the text at lines 29-35 and 289-291. We also improve the description of seismically induced sinkholes in Italy by referring to all the suggested papers in the discussion section, see changes at lines 50-54.

**# Reviewer 1.**

- the few number of independent sources (SBAS, historical data, field survey, and seismic analysis);

their variable quality levels (field observations not enough extended);

- their limited nature (there is no geomorphological map, no structural map, no trenching, no SBAS field validations, no sub-surface geophysics, no boreholes);

- structural map is not presented while this source of data is interesting to link the genesis of the depression with a possible seismic creep;

- geomorphological maps, at local and regional scales, should be drawn in order to confirm that this depression is really a singularity in the landscape. There is no evidence anywhere that this depression is an isolate case or that similar phenomena can be observe elsewhere in the region. It is really important to clarify the status of this de-pression because if it is an isolated case, then, it can be considered a very interesting indicator regarding the tectonic activity in the region;

- the stratigraphy is very poorly described and the thickness of the different layers below the depression is incomplete. A carbonate layer is mentioned in the text (Tuscan Nappe Unit) but not its depth while this layer is a good candidate to be the siege of dissolution phenomena leading to ground subsidence at the surface.

**# Reviewer 2, specific comment 2.1:**

The paper lacks essential data on the geomorphic context, including a detailed map. The latter may show the presence of landslides or other sinkholes in the area. A thorough geomorphological analysis is needed to identify the active processes in the study area and distinguish their relative importance in the sinkhole deformation dynamic. Such as: detailed mapping, trenching combined with geochronological data (to study the geological record and increase the temporal registry), and geophysics.

**Response:** We agree with the reviewers that a better description of the stratigraphy, geology, tectonic structures, and geomorphology of the Pieve Fosciana area is needed to improve the paper, providing more compelling evidences of the presence of an active sinkhole and its relationship to tectonics. Therefore, we added a geomorphological and structural map (Fig. 2a) as well as a geological cross-section of the study area (Fig. 2c). The latter shows the stratigraphy of the Prà di Lama sinkhole and the presence, at  $\sim 2$  km depth, of carbonatic and evaporitic formations. The geomorphological map also shows the presence of the Prà di Lama sinkhole and the lack of landslides in the area affected by ground motion as identified by InSAR. The Prà di Lama sinkhole is an isolated feature in the region being the only mapped sinkhole in the entire Garfagnana graben (Caramanna et al., 2008); the closest sinkholes are near the Tuscany coast. We added to figure 1 the locations of the sinkholes to clearly show the Prà di Lama site is an isolated case. We also expanded on the field observations by adding photos of fractures and cracks around the lake that further document the deformation activity at Prà di Lama (Supp. Fig. 1 and Suppl. Fig. 2). Observations from a 200 m-deep borehole have been also added, showing the detailed stratigraphy (Fig. 2b). A recent study from Gherardi and Pierotti (2018), expanding on previous research (Baldacci et al., 2007), uses geo-chemical analyses of the Prà di Lama spring waters and concludes that the high salinity (~5.9g/kgw) and temperature (~57 °C) are explained by hydrothermal circulation be-tween 1.3 and 2 km depth in an evaporitic-carbonate reservoir. The results from this study are in agreement with the presence of a deep sinkhole at Prà di Lama, and the evaporitic-carbonate reservoir likely corresponds to the anhydrite of the Burano Fm. and the calcareous-dolomitic breccia of the Calcare Cavernoso Fm., as shown by our geologic cross-section Fig. 2c). More importantly, the un-mixing of deep waters in shallow aquifers is interpreted by Gherardi and Pierotti (2018) as an evidence of the rapid upwelling of deep waters along the main tectonic structures (Baldacci et al., 2007). The conclusions reached by the authors have been added to the manuscript in the discussion section. Our revised dataset now includes observations from completely independent methods spanning surface deformation (InSAR), structural geology, seismicity, geomorphology, borehole stratigraphy and a century-long historical record of sinkhole activity. We now also include a comparison of our results to independent geochemistry results. Several independent lines of evidence that support our conclusions are now provided. Data from subsurface geophysics and trenching in Pieve Fosciana are not available and their high cost prevents us to obtain these datasets at present. While these data provide a more detailed view of the source geometry and fault activity, we believe that the main conclusions reached by our study are still supported and our dataset adds new and relevant information to the debate about sinkhole formation and their link to active tectonic structures. We hope that our study will raise the scientific interest in the area and that subsurface geophysics and trenching will be carried out in the future.

**# Reviewer 1.**

Partial temporal and spatial overlap. (most information are concentrated in the last two decades);

- The historical record is a too limited set of observations. They are informative but could become much more relevant if they were complemented by trenching and dating as it is done in paleo-seismology in combination with historical data does not allow a clear understanding of the sinkhole formation. The idea of seismic creep seems to me not supported by a robust analysis performed at local and regional scales. The sub-surface geophysical facet is missing and therefore it is very difficult to be convinced with this explanation. Much deeper investigations are still needed.

Response: The first information about the Prà di Lama lake date back to the 991 A.D., when it was reported that a depression filled by a lake formed from a series of previously isolated springs. We added this explanation to the manuscript to clarify that the historical record allows us to define the time-scale of sinkhole formation. Furthermore, the historical record shows well the episodic behaviour of the sinkhole; this has also been clarified in the manuscript. We agree with the reviewer that the historical record is limited and therefore we complemented it with InSAR, seismicity and structural geology data that have now been expanded to include geomorphology, borehole stratigraphy and a comparison to a recently published geochemistry study. See also our previous response. Regarding the seismicity analysis, we used a reputed catalogue, the Italian national catalogue of seismicity recorded by the Istituto Nazionale di Geofisica e Vulcanologia (INGV) and available online (http://cnt.rm.ingv.it/search). We already analysed the seismic moment release and magnitude contents both in the broader tectonic region and at the local scale. The complete dataset containing all the low magnitude events (MI

**Comment 2.2:** I believe the sinkhole definition used (lines 29-30) is inadequate since not all the sinkholes form due to cavity collapse. There are other genetic processes. The authors should clearly indicate the type of sinkhole they are investigating, explaining the subsidence mechanisms in relationship with the local stratigraphy. I consider that revising this paper: Parise, M., Closson, D., Gutiérrez, F. et al. Environ Earth Sci (2015) 74: 7823. https://doi.org/10.1007/s12665-015-4647-5; could help. The cover is underlain by flysch. Do you have deep-seated caprock collapse sinkholes?

**Response:** We modified the sentence "Sinkholes are quasi-circular depressions in the ground surface that form due to the breakdown of subterranean cavities" to "Sinkholes are closed depressions with internal drainage typically associated with karst environments", following the definitions by *Ford and Williams (2007)* and *Gutierrez et al. (2008, 2014)*. We than added a brief description of the main genetic processes, as suggested by the reviewer, and clarified that we study a sinkhole classified as deep-sited caprock collapse sinkhole, according to *Gutiérrez et al. (2008, 2014)*. We explained the subsidence mechanisms in relationship with the local stratigraphy by clarifying that collapse of deep cavities caused by fluid circulation occurs in carbonatic-evaporitic formations located at 1.3-2 km depth and covered by a thick non-carbonatic sequence. See changes at lines 29-49.

**Comment 2.3:** The authors conclude that "a source mechanism for the sinkhole formation and growth is seismic creep in the active fault zone underneath the sinkhole". Although this hypothesis looks innovative, it is not well supported by the presented data. The casual relationship between creep tectonic deformation and sinkhole activity remains as an unproved hypothesis. I encourage the authors to add sub-subface geophysical and structural data to test their hypothesis.

**Response:** We agree that the explanation of a tectonic-induced sinkhole is new and we provided a new structural geology map (Fig 2a) as well as a geology cross-section (Fig. 2c) showing that faults geometries are consistent with a structural control on the sinkhole. We also added a discussion section including examples of active faults characterized by seismic creep analogous to our case. In particular we presented

the examples of Taiwan (*Rau et al., 2007; Chen et al., 2008; Harris, 2017*) and Parkfield (California) (*Nadeau et al., 1995; Harris, 2017*).

**List of relevant changes**

- Line 3. Prof. Giacomo D'amato Avanzi (Dipartimento di Scienze della Terra Università di Pisa) has been added as co-author of the paper for his contribution of geomorphological information in Pieve di Fosciana.
- Lines 29-32. The sentence "Sinkholes are quasi-circular depressions in the ground surface that form due to the breakdown of subterranean cavities" has been modified to "Sinkholes are closed depressions with internal drainage typically associated with karst environments", following the definition by Ford and Williams (2007) and Gutierrez et al. (2008, 2014)
- Lines 33-49. A better description of the several types of sinkholes has been provided following the classification of *Gutierrez et al. (2008, 2014)* and by referring to *Del Prete et al. (2010) Parise et al., (2010),* as suggested by the editor
- Lines 50-54. A more complete review of Italian seismically-induced sinkholes has been provided by referring to *Parise and Vennari (2013), Parise et al. (2010), Kawashima et al. (2010), Santo et al. (2007),* as suggested by the editor.
- **Lines 88-90.** The stratigraphic sequence of Prà di Lama lake has been completed by describing the deeper formations.
- **Lines 91-95.** We added a description of the geomorphological features characterizing the Prà di Lama lake.
- Lines 100-109. The geochemical analyses of the Prà di Lama springs and the related conclusions provided by *Gherardi and Pierotti (2018)* have been added to our manuscript to better constrain our hypotheses.
- Lines 161-166. We have added a section describing the role of the coherence threshold in the InSAR P-SBAS processing.
- **Lines 174-181.** We have added a section to describe the selection of the reference point and its effect on the final results
- Lines 189-203. We have added a section showing the results of the processing test using a coherence threshold of 0.4, as suggested by the referees
- Lines 242-249. In the seismicity chapter, we have added a section containing the results of the analysis of the Historical Catalogue (CPTI15) and we have shown the lack of correspondence between strong historical earthquakes and events of unrest at Prà di Lama
- **Lines 258-267.** We considered other geological processes that could explain the observed deformation at Prà di Lama, as suggested by the referees

- **Lines 283-286.** We included the geochemical analysis of spring waters (*Gherardi and Pierotti, 2018*) in our discussion.
- Lines 286-288. We classified the Prà di Lama sinkhole as a deep-sited caprock collapse sinkhole using the classification of *Gutierrez et al. (2008, 2014),* as suggested by the editor
- **Figure 1** has been modified to show that the Prà di Lama sinkhole is an isolated feature in the region being the only mapped sinkhole in the entire Garfagnana graben. In Particulare, a red dot has been added to indicate the Camaiore sinkhole and a yellow star has been used to indicate the Prà di Lama sinkhole.
- Figure 2 This is a new figure. It consists of a geological, geomorphological and structural map (Fig. 2a) accompanied by a shallow stratigraphic log from *Chetoni (1995)* (Fig 2b) and a geological cross-section (Fig 3).
- **Figure 3** has been modified to better show the fracture pattern formed during the unrest event of 1996.
- **Figure 8** has been added to show the comparison between the historical seismic catalogue and the events of unrest at Prà di Lama.
- **Supplementary material** has been included. Supplementary material 1 and 2 contain pictures of the two recent events of unrest at Prà di lama (1996, 2016). Supplementary material 3 show the InSAR processing results using a coherence threshold of 0.4.

- 1 GROWTH OF A SINKHOLE IN A SEISMIC ZONE OF THE NORTHERN APENNINES (ITALY)
- Alessandro La Rosa1,2, Carolina Pagli2, Giancarlo Molli2, Francesco Casu3, Claudio De Luca3,
   Amerino Pieroni4 and Giacomo D'amato Avanzi2
- 4

[revised manuscript text omitted]

- 529
- 530
- 531
- 532
- 533
- 534
- 535

---

## Editor Decision (ED1)

[revised manuscript text omitted]

---

## Author Response (AR2)

**Reply to the comments and List of relevant changes**

We thank the editor for these new comments. All the suggested changes have been incorporated in the revised version of the manuscript. Here it follows the list of the relevant changes.

- References have been listed in chronological order, as suggested, at lines 36, 46, 51, 94, 269, 272, 275, 290.
- A new reference about the Camaiore sinkhole has been included in the new version of the manuscript at line 91 and in figure 1. We referred to *Buchignani, V., D'Amato Avanzi, G., Giannecchini, R., Puccinelli, A.: Evaporite karst and sinkholes: a synthesis on the case of Camaiore (Italy). Environmental Geology, 53, 1037-1044*. https://doi.org/10.1007/s00254-007-0730-x*, 2008.*
- We better described the vegetation at Prà di Lama and its impact on the coherence loss at lines 191-193
- We provided further information about the book "Pieve Fosciana Ieri e Oggi" by adding the "Amministrazione Comunale di Pieve Fosciana" as editor. We also added the place and the number of pages.

[revised manuscript text omitted]